# Photon-directed multiplexed enzymatic DNA synthesis for molecular digital data storage

Howon Lee[1,2], Daniel J. Wiegand [1,2], Kettner Griswold[1,2,3,4], Sukanya Punthambaker [1,2], Honggu Chun[5], Richie E. Kohman[1,2 ✉] & George M. Church [1,2 ✉]

New storage technologies are needed to keep up with the global demands of data generation. DNA is an ideal storage medium due to its stability, information density and ease-of-readout with advanced sequencing techniques. However, progress in writing DNA is stifled by the continued reliance on chemical synthesis methods. The enzymatic synthesis of DNA is a promising alternative, but thus far has not been well demonstrated in a parallelized manner. Here, we report a multiplexed enzymatic DNA synthesis method using maskless photolithography. Rapid uncaging of $Co^{2+}$ ions by patterned UV light activates Terminal deoxynucleotidyl Transferase (TdT) for spatially-selective synthesis on an array surface. Spontaneous quenching of reactions by the diffusion of excess caging molecules confines synthesis to light patterns and controls the extension length. We show that our multiplexed synthesis method can be used to store digital data by encoding 12 unique DNA oligonucleotide sequences with video game music, which is equivalent to 84 trits or 110 bits of data.

[1] Department of Genetics, Harvard Medical School, Boston, MA 02115, USA. [2] Wyss Institute for Biologically Inspired Engineering, Boston, MA 02115, USA. [3] MIT Media Lab, Massachusetts Institute of Technology, Cambridge, MA 02139, USA. [4] Charles Stark Draper Laboratory, Cambridge, MA 02139, USA. [5] Department of Biomedical Engineering, Korea University, 466 Hana Science Hall, 145 Anamro, Seongbukgu, 02841 Seoul, South Korea. ✉email: richie.kohman@wyss.harvard.edu; gchurch@genetics.med.harvard.edu

In the era of Big Data, molecular DNA has become an increasingly attractive medium for the storage and archiving of digital data[1–6]. This is primarily attributed to its ultra-high storage density, which is currently estimated to be in the hundreds of petabytes per gram DNA[7]. The attractiveness of DNA as a data storage medium is additionally bolstered by its durability, longevity, and energy efficiency compared to counterpart storage mediums; both analog and digital[8–11]. However, for storing a meaningful volume of digital data, the synthesis of many unique DNA sequences at long lengths is required. While advances in array-based Oligonucleotide Library Synthesis (OLS) technology have enabled highly multiplexed DNA oligonucleotide synthesis, production in this format still relies on decades old phosphoramidite chemical synthesis methods[9]. Many time-consuming steps of expensive and harsh reactions with the accumulation of toxic by-products greatly limit chemical synthesis as the demand for longer and larger quantities of DNA oligonucleotides increases[10].

Recently, the use of terminal deoxynucleotidyl transferase (TdT), a template-independent polymerase, to synthesize DNA oligonucleotides was shown to be a promising alternative to chemical synthesis[11–13]. Because synthesis reactions are performed under aqueous conditions, many limiting aspects of the phosphoramidite chemistry can be circumvented or improved upon. This would be ideal for digital data storage in DNA; however, due to the natural promiscuity of TdT, controlling the enzyme in a sequence-specific manner is challenging[14–16]. In order to overcome this challenge, a recent study showed controlled TdT extension activity with apyrase by degrading free nucleotides needed for synthesis[13]. Incubation with optimized ratios of apyrase, TdT, and nucleotide over multiple cycles resulted in the successful synthesis of several DNA oligonucleotides comprised of short homopolymeric blocks encoding digital information within unique base transitions. While there is a clear potential for enzymatic-based methods, its widespread adoption for applications such as digital data storage is hindered by the lack of parallelized synthesis technologies. Here, we demonstrate an enzymatic DNA oligonucleotide synthesis platform that utilizes photolithography to selectively modulate TdT extension activity in a multiplexed array format. We show that our multiplex synthesis method can be used to store digital data by encoding 12 unique DNA oligonucleotides sequences with 84 trits (110 bits) of video game music in non-identical base transitions. To recover digital data without previous knowledge of template sequence composition or use of error correct bits, we utilize physical sequence redundancies, a stringent filtering process and stochastic estimation.

## Results

**Parallelized enzymatic DNA synthesis**. The photolithographic activation of TdT is achieved by optically controlling the 'spatiotemporal' concentration distribution of its metal cofactor cobalt ($Co^{2+}$) near the array surface with the photocleavable caging molecule DMNP-EDTA[17–19]. Initially, all $Co^{2+}$ ions are complexed with DMNP-EDTA, which is provided in excess, keeping TdT in an inactive state until ready for synthesis. Upon photolysis of complexed DMNP-EDTA with patterned UV light, $Co^{2+}$ ions are uncaged resulting in a localized pulse of cofactor concentration and spatially selective oligonucleotide extension by TdT within the defined areas of irradiation. To control the number of extension events, the UV light source is turned off after a pre-determined time allowing excess DMNP-EDTA to diffuse into irradiated areas, chelating $Co^{2+}$ ions and returning TdT to an inactive state. This sequence of events constitutes a full cycle of synthesis (Fig. 1a). Virtually any individual position on the array

surface is spatially addressable by our system using a computer-controlled spatial light modulator, or digital micromirror device (DMD), to facilitate the on-demand generation of dynamic light patterns without needing to construct physical masks (Fig. 1b). This functionality is paramount for controlled multiplexed oligonucleotide synthesis on the array surface.

We initially found that several key system parameters required optimization before we could perform template-independent DNA oligonucleotide synthesis in multiplex. First, irradiation of the surface with UV light at an energy density of at least 2 J/cm$^2$ (2 W/cm$^2$ for 8–15 s) was critical for the sufficient uncaging of $Co^{2+}$ ions and TdT activation. Lower UV doses resulted in little observable enzymatic activity as characterized by very weak or no surface signal whatsoever. This suggested that either incomplete photolysis of DMNP-EDTA occurred or that the uncaging rate of $Co^{2+}$ ions was significantly lower than the rate at which they were spontaneously chelated by free DMNP-EDTA. Supporting evidence for the latter was given by the necessity to finely tune the total concentration of DMNP-EDTA initially supplemented in the synthesis reaction master mix.

We determined that the DMNP-EDTA concentration needed to be at least 1.3× that of $Co^{2+}$ for proper spatial confinement on the surface; however, this was also largely dependent on the fill factor of the pattern generated by the DMD (Supplementary Fig. 1). We also found that a post-illumination incubation for 15–20 s significantly boosted the overall yield of oligonucleotide extension on the surface (Supplementary Fig. 2). This incubation period allowed TdT enough time to incorporate nucleotides upon $Co^{2+}$ ion uncaging before being quenched by DMNP-EDTA when the UV light source was turned off. The enzyme concentration and percentage of glycerol in the master mix also affected the kinetics of nucleotide incorporation but were overall less impactful (Supplementary Fig. 3). All reactions were performed at room temperature rather than 37 °C for user convenience and to prevent excessive extension by TdT.

While optimization of our system was multifactorial, we ultimately decided to use a mask pattern of 100 μm diameter circular spots arranged in a (3 · 4) array on 1.2 mm$^2$ of surface area. This yielded 12 individually addressable locations for multiplexed oligonucleotide synthesis (Fig. 2a). To confirm that these conditions were suitable for proper extension, we synthesized homopolymers consisting of only G at all 12 spots from the 3′-terminus of short initiator sequences anchored to the surface. Positive extension was verified using fluorescent imaging after the ligation of a short visualization probe containing a Cy3 fluorophore using a sequence-specific splint (Supplementary Fig. 4a and Methods section). No extension outside the illuminated areas or erroneous crosstalk between the 12 spots was observed on the surface, indicating robust nucleotide incorporation by TdT and that DMNP-EDTA quenching functioned appropriately to spatially confine DNA oligonucleotide synthesis (Fig. 2b).

**Multi-cycle synthesis**. We next sought to normalize the rate of incorporation for all nucleotide types by TdT in preparation for multi-cycle oligonucleotide synthesis as it is well-known that they can be highly variable given the composition of the initiator sequences' last 3–5 bases[15]. To do this, we increased or decreased the total UV illumination time on the surface for a given base transition. Using the same 12 spot pattern, we empirically found that base transitions involving initiator oligonucleotides ending with G or C required longer illumination times (>12 s) as compared to those ending with A or T (≤10 s) for all nucleotide types (Fig. 2c and Supplementary Fig. 5). The longest illumination time was 17.5 s for the base transition C to T, while the shortest was 6 s

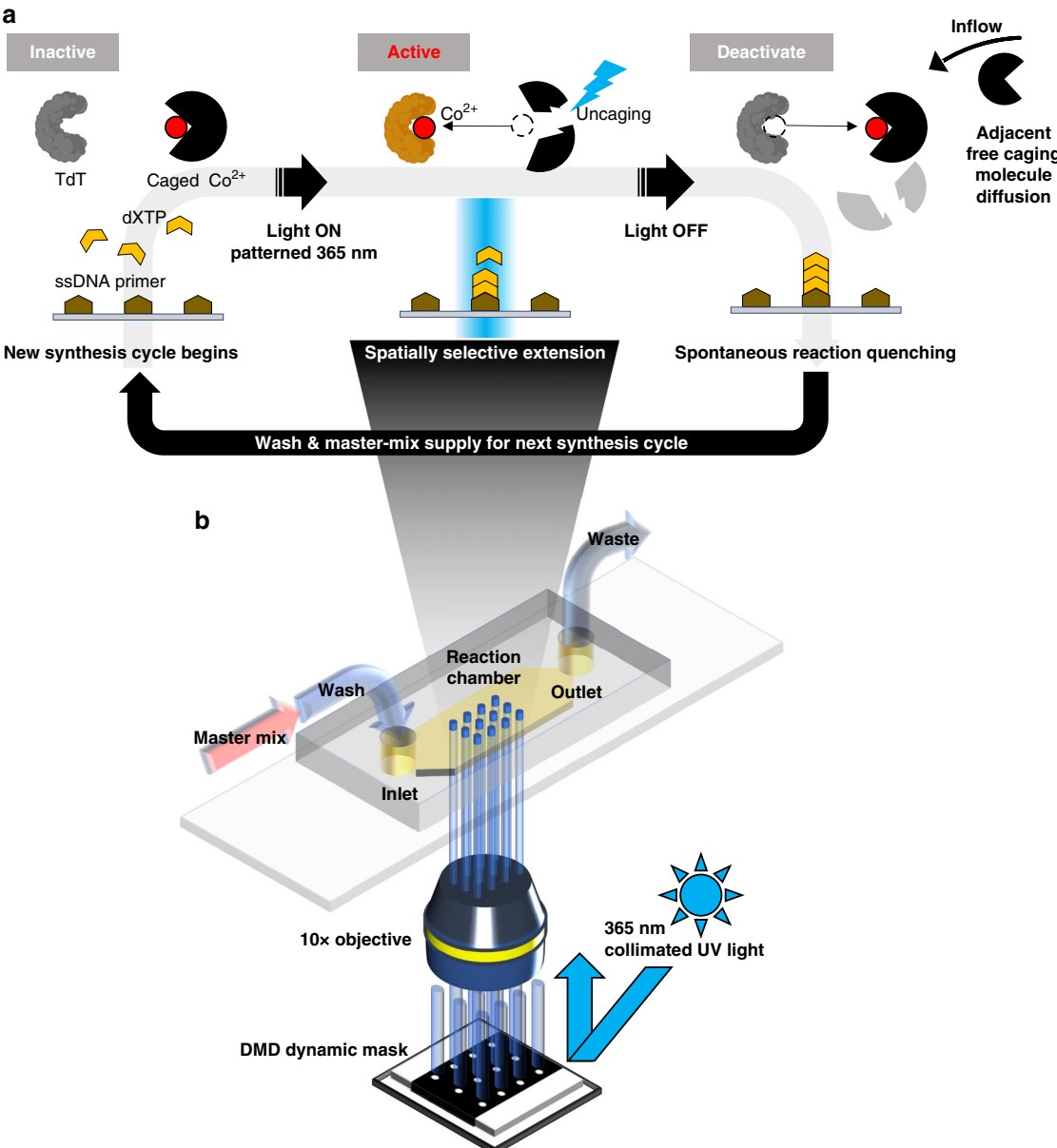

**Fig. 1 Overview of photon-directed multiplexed enzymatic DNA synthesis system. a** An array surface derivatized with single-stranded DNA initiator oligonucleotide is brought into contact with a master mix containing the appropriate buffers, $Co^{2+}$ divalent cation cofactor, TdT enzyme, the desired nucleotide to be incorporated (dXTP), and photolabile DMNP-EDTA caging molecule provided in excess. All $Co^{2+}$ ions are initially complexed with DMNP-EDTA, which causes TdT to remain in an inactive state until needed. Using photolithography, patterned UV light at 365 nm illuminates the array's surface causing the complexed DMNP-EDTA to degrade releasing $Co^{2+}$ ions and activates TdT in a spatially selective manner. The UV light is then turned off and the reaction is allowed to incubate for a short period of time. During this incubation, excess, non-complexed DMNP-EDTA spontaneously quenches the extension reaction by chelating free $Co^{2+}$ causing active TdT to become inactive. The array surface is then washed and either the next synthesis cycle begins or material is retrieved from the surface for downstream applications. **b** Arrays are mounted into a simple flow cell with a reaction chamber inlet and outlet to waste or collection. Individually addressable patterning is a major advantage of our synthesis method, which is provided by the generation of reflective on-demand dynamic masks from the DMD through a ×10 objective from a collimated UV light source.

for A to G. Based on this observation, the standard UV exposure protocol for multiplexed synthesis was determined as two normalized illumination times (Methods section). For all synthesis spots ending with C in any synthesis cycle or step will be illuminated 20 s when they newly incorporate non-identical nucleotides at the 3′ end, otherwise the UV exposure time was set to be 10 seconds uniformly.

Because these results only indicated a relative comparison of overall TdT activity for each condition, we employed next-generation sequencing (NGS) to determine the actual number of

extension events that occurred for each base transition. We again utilized our sequence-specific splint-end ligation approach to ligate the appropriate adapters for PCR amplification and NGS library preparation. With this, only those sequences that were properly and robustly extended would be extracted for NGS analysis (Supplementary Fig. 4b). Overall, we found that the normalization of illumination times resulted in an average of 4 to 8 extension events for all tested base transitions; however, >15 extension events were possible, but were only observed in a fraction of what was synthesized (Fig. 2d).

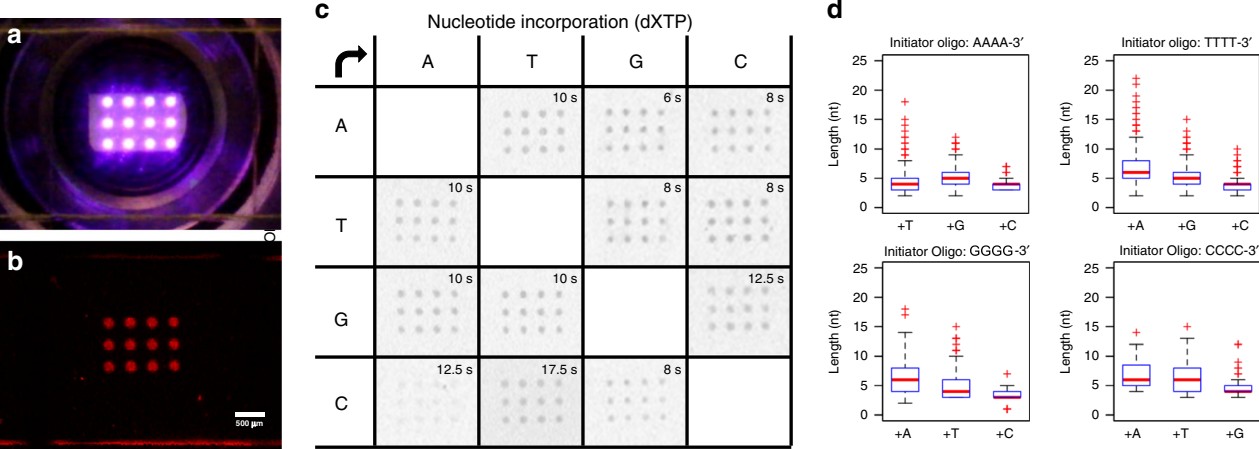

**Fig. 2 Multiplex enzymatic synthesis optimization and base transition normalization. a** Demonstration of UV irradiation of the array surface with 100 μm circular spots arranged in a (3 · 4) patterned format on 1.2 mm² of surface area. UV irradiation is not limited to this particular patterning and may be pixel-wise (1920 × 1080) changed on-demand with our photolithographic system. Any spot on the surface is individually addressable in terms of spatial location and the total amount of UV irradiation time. **b** Visualization of G homopolymeric oligonucleotide synthesis post system optimization via the splint-end ligation of a probe sequence containing a 3′-Cy3 fluorophore using the (3 · 4) pattern. **c** Results of base transition normalization in which the total illumination time was adjusted for any base transition that may be encountered during multiplexed synthesis. The left axis indicates the composition of the last 4 bases on the 3′-terminus of surface initiator oligonucleotide and the top axis indicates the nucleotide that was incorporated onto the respective initiator oligonucleotide. The optimal illumination time is indicated in each base transition box and was determined by splint-end ligated 3′-Cy3 fluorescent signal. **d** Box plots indicating NGS analysis of base transition normalization. Graphs show the normalized nucleotide (nt) extension length distribution for all possible base transitions with red pluses being statistical outliers. Source data are provided as a Source Data file.

We next sought to prove that our system was capable of multi-cycle synthesis by producing a heteropolymeric sequence consisting of all four natural nucleotide bases in single-plex on the (3 · 4) array (Fig. 3a). We elected to synthesize the 8 base transition DNA sequence GATGTAGAC with the expectation that a successful demonstration would yield oligonucleotide comprised of short homopolymeric blocks for each base used, similar to previous work[13]. Synthesis began with an anchored initiator oligonucleotide consisting of a 3′-string of four G and ended with a final base transition of C for adapter ligation and sequencing purposes. Enzyme master mixes supplemented with the appropriate nucleotide were applied to the flow cell in the designated order with all 12 spots illuminated during each cycle to produce the target sequence. Following the completion of eight synthesis cycles, we verified that the correct sequence was produced and determined the extension length distribution for each homopolymer block using NGS as previously outlined (Fig. 3b and Methods section). We additionally tracked the progress after each cycle by performing parallelized synthesis across several flow cells with which we could then analyze TdT extension with gel electrophoresis after retrieval from the surface (Fig. 3c). In both cases, the distribution of block lengths was in line with what we expected post-normalization of the illumination times.

Having determined that multi-cycle synthesis was possible in single-plex, we next turned to showing that the same process could be applied in multiplex in order to produce 12 unique oligonucleotide sequences on the array surface simultaneously (Fig. 3d). We accomplished this by taking advantage of the ability for our system's DMD to rapidly generate dynamic mask patterns on demand. This allowed us to incorporate nucleotide bases at any particular spot or combination of spots on the (3 · 4) array. As an added benefit, individual spot illumination time could be adjusted to meet the established TdT normalization criterion for any given base transition (Fig. 2c, d). A total of 29 unique masks and incorporation cycles over eight steps of synthesis were required to produce the desired oligonucleotides in multiplex

(Supplementary Fig. 6). One step of multiplex synthesis contained the incorporation cycles for all four natural nucleotides, which could be algorithmically filtered and individually analyzed for homopolymeric block length post-sequencing (Fig. 3e). If a step did not require one or more bases, those cycles were skipped by not illuminating the surface.

**Data encoding, decoding, and error analysis.** To demonstrate that our synthesis method can be used for digital data storage applications, the 12 oligonucleotide sequences synthesized in multiplex encoded the first two measures of the "Overworld Theme" sheet music from the 1985 Nintendo Entertainment System (NES) video game Super Mario Brothers (Supplementary Fig. 7a, b). Our encoding and decoding scheme was based on previous work[13], where digital data is stored into the homopolymeric blocks of unique base transitions produced by TdT (Fig. 4a–g). Input musical information was indexed, assigned a note number based on a Musical Instrument Digital Information (MIDI) chart, converted to ternary, and from this, the 12 unique DNA oligonucleotide sequences were generated (Fig. 4a–c and Supplementary Figs. 8a–c and 9). In addition to sheet music, which contained information regarding the notes as well as the order and duration each note is to be played, we used the first three base transitions to store the physical locations of each oligonucleotide on the (3 · 4) array as barcodes. In total, this amounted to 84 trits, or 110 bits, of data stored digitally in the 12 DNA oligonucleotide sequences.

Following multiplex synthesis and NGS, filtered sequence data was algorithmically decoded and processed by a sinusoidal wave generator to play the sheet music in true sound. In silico filtering occurred in three discrete stages and was carried out to ensure that sequences synthesized in multiplex could be recovered without previous knowledge of their composition after fulfilling a set of predetermined initial boundary conditions (Fig. 5a, Supplementary Fig. 12, and Methods section). Briefly, we first selected reads that contained the correct initiator oligonucleotide sequence and splint-ligated sequencing adapter. From these reads,

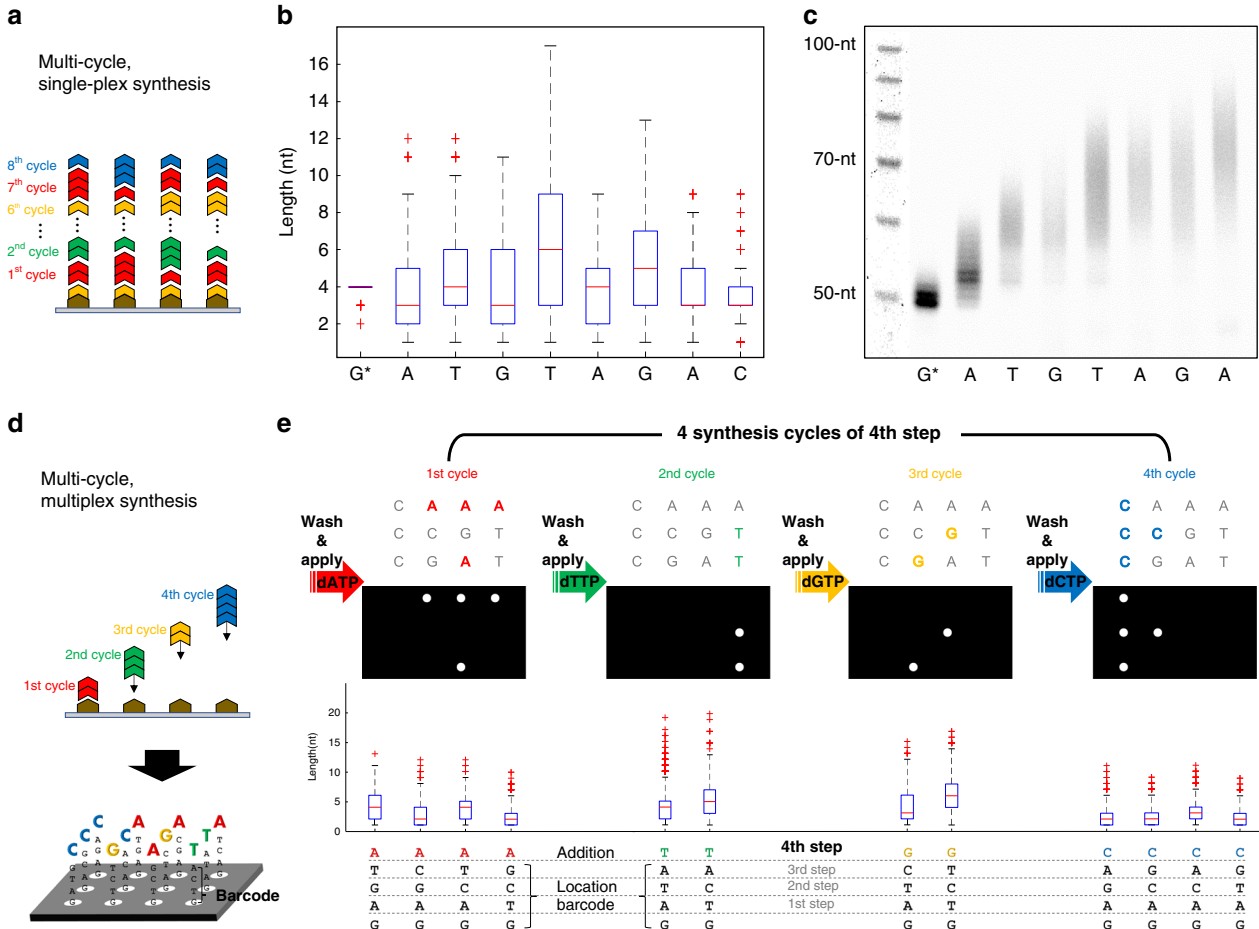

**Fig. 3 Demonstration of multi-cycle and multiplex enzymatic synthesis. a** An overview of multi-cycle, single-plex synthesis of a heteropolymer oligonucleotide comprised of 8 unique base transitions. Each cycle represents the addition of a single-nucleotide base type at all 12 spots on the (3 · 4) array. Synthesis results in short homopolymeric blocks of A, T, G, or C at variable lengths. **b** Following multi-cycle synthesis and NGS, raw sequence reads were extracted and filtered by the presence of the adapter sequence added by splint-end ligation after the final C extension. The box-plot indicates a statistical representation of the number of extension events for each homopolymeric block for the synthesized sequence GATGTGAGAC with red pluses being statistical outliers. Note that only sequencing reads that contained all eight base transitions were used to generate the box-plot and that synthesis started with a string of four G on the 3′-terminus of the initiator oligonucleotide. **c** Denaturing gel electrophoresis analysis of each individual cycle of synthesis. Each lane represents the material retrieved of an individual flow cell after the appropriate number of cycles were performed. No final C extension or splint-end ligation occurred for this analysis. **d** An overview of multi-cycle, multiplex synthesis of 12 unique heteropolymer oligonucleotides on the (3 · 4) array. Each synthesis step contained the individual cycles for the addition A, T, G, and C at the appropriate spots on the array. Sequence barcodes indicating the physical location of each oligonucleotide on the array can be built into the initiator oligonucleotide or synthesized by TdT. For each cycle, a unique mask was generated by the system's DMD to locally activate TdT based on the desired sequence to be synthesized. **e** For example, the masks needed for the 4th step of an eight-step multiplex oligonucleotide synthesis run is shown with the appropriate post-synthesis sequencing data to verify spatially selective synthesis. Source data are provided as a Source Data file.

we subsetted those with 1 of the 12 index barcodes and then selected for sequences that indicated exactly eight full base transitions within the individual subsets. At this point, stochastic estimation indicated that the dominant population of each subset yielded the expected oligonucleotide sequences and they can be decoded to digital music data without needing to implement additional error correction bits (Fig. 5c and Supplementary Figs. 10a–c and 11a, c). A fourth filter was then used to determine the effectiveness of stochastic estimation by comparing the composition of the dominant sequences to their reference templates. We observed only a small reduction in total remaining sequencing reads after the fourth filter indicating that the series of filtering process followed by stochastic estimation was largely successful (Supplementary Fig. 12). In addition, we found that each subset contained the expected normalized distribution of homopolymeric block extension lengths. This was also observed across the entire (3 · 4) array

for all possible base transitions during multiplex synthesis (Fig. 5d and Supplementary Fig. 11b).

An appreciable percentage of the filtered reads contained errors; however, this did not hamper our ability to recover the correct oligonucleotide sequences from the sequencing data due to the physical redundancies built into our system. Errors could have arisen from insufficient activation of TdT, poor washing of the flow cell, and to much lesser extent, unwanted crosstalk between illumination spots. To characterize them, we categorized errors as either base transition insertions, deletions, or mismatches. Single base deletions were found to be the most prevalent (25.8%), while a lower chance of single base insertions (13.4%) or mismatches (8.9%) was observed (Fig. 5b). Less than 5% of the filtered reads contained instances of two or more base transition errors. In breaking this down further, we did not find a strong correlation between any particular base transition (e.g. A to T) and error type, although a noticeably high number of reads containing the

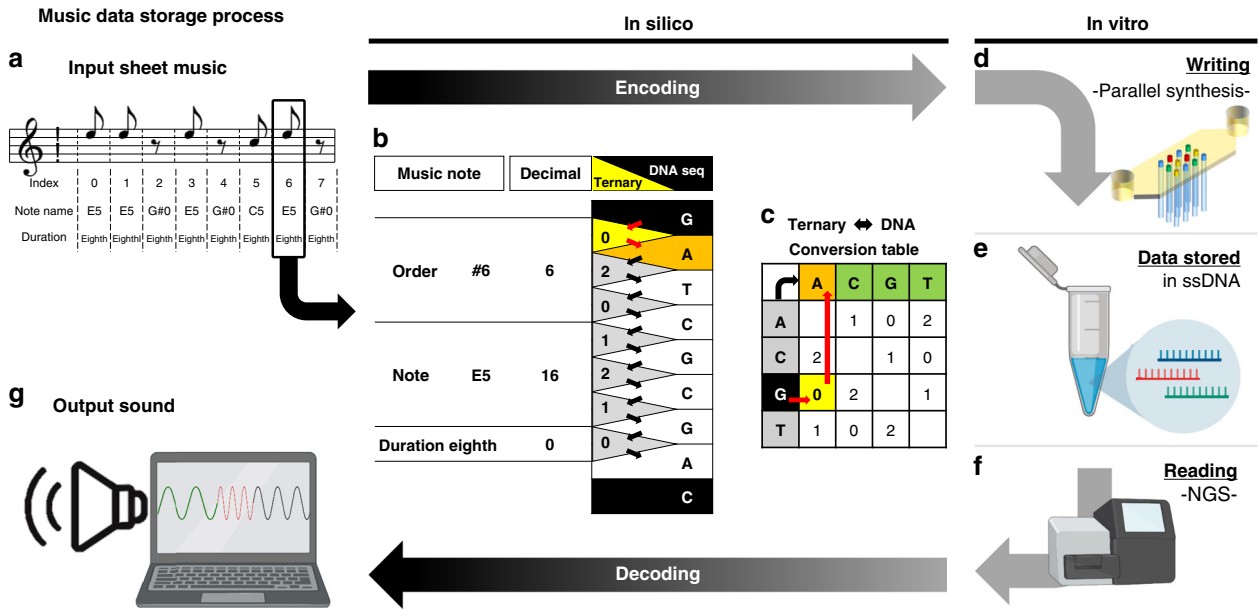

**Fig. 4 Demonstration of digital music data storage in DNA oligonucleotides enzymatically synthesized in multiplex: music data storage process. a** A snapshot of the simplified melody from the two first measures of the 1985 Nintendo Entertainment System video game Super Mario Brothers "Overworld Theme" piano sheet music. Each note and rest of the melody was indexed as #0 through #11 and stored into one of the 12 unique DNA oligonucleotides synthesized in multiplex on the (3 · 4) array. The full piano sheet music is shown in Supplementary Fig. 7. **b** Indexed notes and rests were assigned a note number based on a modified Musical Instrument Digital Information note chart, which also indicates note octave. In addition to the note, the encoding scheme also stores the order and duration each note is to be played. All rests are assigned to the note G#$_0$, which is inaudible (25.9 Hz) to normal human hearing[25]. A full overview of MIDI conversion of sheet music to digital data is shown in Supplementary Fig. 8. **c** Digital music data is translated to ternary and the unique DNA sequences are mapped to unique base transitions using a conversion map. The left axis of the table represents the string of bases at the 3'- terminus of the initiator oligonucleotide and the top axis indicates the nucleotide to be incorporated next. Supplementary Figure 9 indicates a table outlining the mapped "Overworld Theme" musical melody to DNA sequences with all relevant data and information. **d** Generated DNA sequences are enzymatically synthesized with TdT using our multiplex photolithographic system. **e** Synthesized oligonucleotides are retrieved from the flow cell surface and stored in tubes for sequencing or other downstream applications. **f** To read and decode stored digital music data, DNA oligonucleotides synthesized in multiplex can be sequenced with NGS such as Illumina or nanopore methodologies. **g** From sequencing data, the decoding process converts sequencing information back to musical notes. A sinusoidal wave generator is used to play the "Overworld Theme" melody in the correct note order with the proper duration in true sound.

transitions G to C and C to A were prone to error in general (Fig. 5b). This is in line with past observations where most base transitions involving C in some regard proved to be troublesome and required longer illumination times for sufficient enzymatic extension on the surface (Fig. 2c and Supplementary Fig. 5). Nevertheless, introducing logical redundancy such as error correction code[3,13] and reducing the prevalence of base transition errors through further system optimization will be critical for the storage and recovery of high-quality digital data in DNA.

## Discussion

Taken together, our results represent the first demonstration of template-independent, multiplexed enzymatic synthesis of DNA oligonucleotides for digital data storage through the application of photolithography to control enzymatic polymerization. We show that the first two measures of the "Overworld Theme" sheet music from the 1985 NES video game Super Mario Brothers can be encoded digitally into DNA and successfully recovered without previous knowledge of sequence composition despite the accumulation of some base transition errors. Our system utilizes a combination of commercially available photolabile materials and maskless lithography with the added benefit of circumventing the use of expensive physical photomasks, special phosphoramidites, harsh organic solvents, and the accumulation of toxic waste from chemical synthesis. Additionally, since metal cation cofactors are essential for polymerase catalysis, the presented method will be potentially suitable for controlled synthesis using mutated

enzymes with enhanced attributes such as higher processivity or the ability to incorporate modified nucleotides[20].

While enzymatic-based DNA oligonucleotide synthesis technology is still in its infancy, we envision that pairing an overall cleaner, faster and more flexible synthesis methodology with high-density array photolithographic techniques will pave the way for its widespread adoption. However, in order to achieve this, there are still several remaining challenges that need to be first overcome. For example, having the ability to scale easily in terms of both higher levels of multiplexing and the synthesis of longer DNA oligonucleotide sequences (>100-nt) will be essential in promoting the further use of our photon-directed synthesis platform as practical molecular data storage technology. We believe that the synthesis of a greater number of oligonucleotides in multiplex will be enabled by utilizing larger arrays with denser patterning. Because feature size on the array largely depends on parameters such as wavelength, mask resolution, and projection optics, this should be relatively easy to address through the implementation of commercially available high-resolution photolithography systems. Nevertheless, confining the enzymatic reactions to smaller and more dense synthesis spots may require significant optimization and the overall process could be affected in a greater capacity by improper diffusion of DMNP-EDTA, stray light crosstalk or unforeseen inefficiencies in uncaging by patterned UV irradiation[21]. Similarly, achieving longer oligonucleotide synthesis with our system requires the implementation of full automation and may be affected by phenomenon such as the

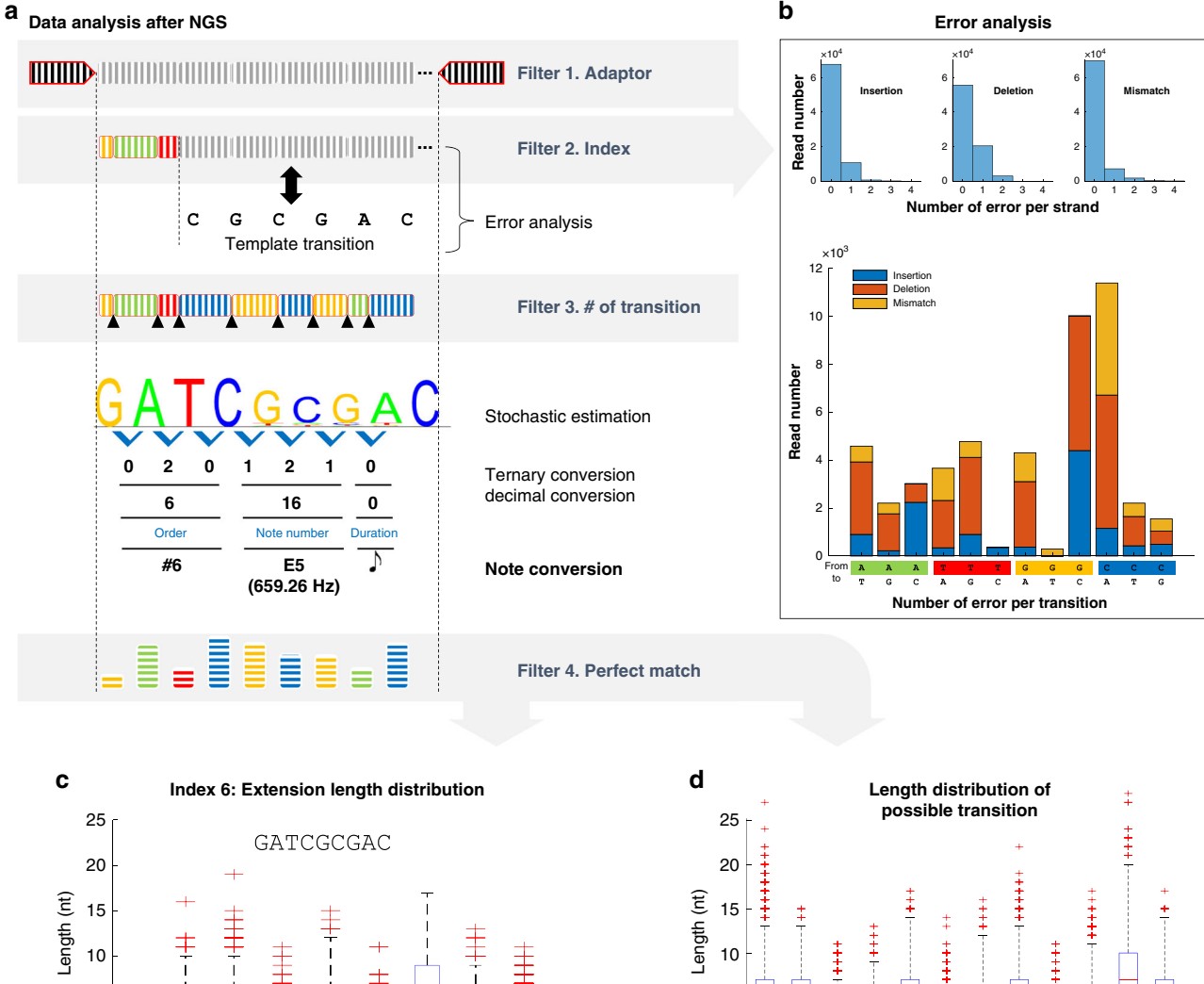

**Fig. 5 Demonstration of digital music data storage in DNA oligonucleotides enzymatically synthesized in multiplex: decoding and data analysis.**
**a** During the decoding process, several filters are applied to extract and align the reads that contain the digital music data to the expected template sequences. Template alignment was performed using the Smith–Waterman algorithm. **b** From this, error analysis can be performed to determine the quality of multiplex synthesis. The upper histogram indicates the percentage of insertions, deletions, or mismatches that occurred in the filtered sequencing reads. The bottom histogram indicates the number of reads containing errors for each possible base transition across the array. **c** Sequencing data also yields statistical information regarding the extension length distribution for each base transition for all 12 oligonucleotides synthesized in multiplex. For example, subset 6, which indicates sequence index 6 is shown in the box plot. All other subsets are indicated in Supplementary Figs. 10 and 11. **d** Additional statistics such as the extension length distribution for all possible transitions from the entire array were analyzed as shown in the indicated box plot with red pluses being statistical outliers. Source data are provided as a Source Data file.

formation of secondary structure that makes the 3′-terminus of the DNA inaccessible by the enzyme without intervention by DNA binding proteins or elevated reaction temperatures[22,23]. This limitation is an inherent problem with aqueous-based reaction systems, which is both well-recognized and currently being addressed with success by several other groups[12,24]. Looking into the future, we believe that efforts to harness multiplexed enzymatic oligonucleotide synthesis in the context of important applications such as digital data storage will usher in a new era of scientific research and biotechnology.

## Methods

**Maskless lithography system and flow cell fabrication.** Our maskless lithography system consisted of a high-power UV LED array with collimation adaptor (LumiBright PR, 2910A-100, Innovations in Optics) couple with Tube lens (MT-L Accessory Tube Lens, Edmund Optics), DMD (DLP4000, Texas Instruments), and Objective lens (CFI Plan Fluor 10X, Nikon) to expose an ultraviolet (365 nm) pattern (Fig. 1a, b). A self-designed computer program in MatLab and control circuit board (Arduino Uno) was used to synchronize DMD patterning and UV illumination time. Flow cells consisted of a cover, spacer, and bottom glass slide with a single inlet and outlet (Fig. 1b). The cover and spacer were fabricated out of acrylic and assembled with double-sided adhesive tape (9172MP, 3M). The inlet and outlet on the flow cell cover and the flow cell pattern of the spacer was precisely cut using a laser cutter (Epilog Legend 36EXT).

**Initiator oligonucleotide surface derivatization.** Streptavidin-coated glass slide (ArrayIt Corporation) was used to anchor 5′-biotinylated initiator oligonucleotide (Integrated DNA Technologies), which served as the bottom of the fully assembled flow cell. To derivatize the streptavidin-coated glass slide, biotinylated oligonucleotide was incubated at a final concentration of 0.25 mM in 1X Binding & Wash

buffer (20 mM, 1 M NaCl, 1 mM EDTA, 0.0005% Triton-X100, and pH 7.5) for 1 h at room temperature. After incubation, the surface was washed with fresh 1X Binding & Wash buffer and then washed again with 1X phosphine buffer saline (PBS). A standard initiator oligonucleotide consisted of the sequence (/5Biosg/ TGGTTAGTGTGCTTCGGACCGGGG) for initial system optimization and final multiplexed synthesis demonstration. For normalized base transition experiments, the initiator oligonucleotide sequence was the same with the exception of the last four bases on the 3′-end of the oligonucleotide. These were variable (either -GGGG, -CCCC, -AAAA, or -TTTT) depending on the target base transition.

**Standard master mix composition.** A standard synthesis master mix was composed of 20 units of recombinant calf thymus TdT enzyme (Thermo Scientific), 1X reaction buffer (0.2 M potassium cacodylate, 0.025 M Tris, 0.01% (v/v) Triton X-100, 1 mM $CoCl_2$), 0.1 mM of either dATP, dTTP, dGTP, or dCTP (Invitrogen), and 1.3 mM caging molecule DMNP-EDTA tetrapotassium salt (MilliporeSigma) in 10 μL of deionized water. All reaction incubations were performed at room temperature.

**Standard synthesis cycle.** A standard synthesis cycle consisted of loading the required mask image to the DMD, delivering 2 μL of synthesis master mix to the flow cell, and then applying UV irradiation to the bottom surface for 10–20 s depending on the base transition occurring at the individual illumination spots (Fig. 2c). For those base transitions that required very short illumination times, a minimum of 10 s of illumination was used and 20 s for longer cases (2-discrete steps). A post-illumination incubation would then take place for at least 20 s to ensure optimal TdT extension and reaction quenching. Synthesis master mix was removed from the flow cell to the waste by vacuum suction and then washed with 20 μL of 1X PBS. Remaining 1X PBS wash was removed from the flow cell to the waste also by vacuum suction. To start the next cycle, a new mask image would be loaded into the DMD and this process was repeated as necessary for each nucleotide to be incorporated during a particular step in the case of multiplex synthesis (Fig. 3a, d, e and Supplementary Fig. 6).

**Sequence-specific split-end ligation.** To visualize oligonucleotide synthesis without removing oligonucleotide from the surface, we employed a sequence-specific splint-end ligation technique that allowed us to add a short oligonucleotide probe containing a 3′- Cy3 fluorophore to the end any oligonucleotide synthesized using our method (Supplementary Fig. 4a, b). Splint-end ligations were performed using a Quick Ligation Kit (NEB) as per manufacturer's instructions. Reactions consisted of 1x Quick Ligase Buffer, 25 μM Cy3 labeled probe (/5Phos/CGA CTG AAC CCA AGC AAC TGA/3Cy3Sp/), 20 μM splint oligonucleotide (CA GTT GCT TGG GTT CAG TCG XXXX, X can be A, G, T, C depending on the synthesized strand), and 1 μL of Quick Ligase in deionized water in 20 μL of volume. In all, 5 μL of the Quick Ligation master mix was delivered to the flow cell and incubated for 2 h at 16 °C. After incubation, the flow cell was washed, disassembled, dried thoroughly with forced air, and imaged using a Typhoon FLA 9000 Imager (GE) with settings for Cy3 fluorescence filter with a 10-μm resolution. This ligation technique was additionally applied to selectively add PCR amplification and NGS adapters to oligonucleotides correctly synthesized.

**Retrieval of oligonucleotides from surface.** To retrieve surface-bound oligonucleotide post-synthesis, a solution of 95% formamide in deionized water supplemented with 10 mM EDTA was applied to the disassembled flow cell bottom surface and heated to 65 °C for 5 min. This solution was then removed from the flow cell bottom surface and cleaved oligonucleotides were purified using a Clean & Concentrator Oligonucleotide Spin Column (Zymo) as per manufacturer's instructions. Oligonucleotides were eluted into deionized water for downstream processing or sequencing.

**PCR amplification and gel electrophoresis analysis.** Oligonucleotides removed from the surface with appropriate adapters for PCR were amplified using a KAPA SYBR Fast qPCR 2X Master Mix Kit (Roche) as per manufacturer's instructions and purified QiaQuick PCR Clean-up columns (Qiagen). Amplified or raw oligonucleotide material was analyzed using 15% TBE-Urea precast polyacrylamide gels as per manufacturer's instructions. Oligonucleotide material amplified with Cy3 labeled primers was visualized using a Typhoon FLA 9000 Imager (GE) with settings for Cy3 fluorescence following gel electrophoresis.

**Illumina MiSeq library preparation & sequencing.** Purified PCR amplified sequences containing encoded data were prepared for next-generation sequencing using a NEBNext Ultra II DNA Library Prep Kit for Illumina as per manufacturer's instructions with ~100 ng of material per library. Libraries were not size-selected during magnetic bead clean-ups in order to preserve the true length distribution of sequences synthesized on the surface of the arrays as best as possible. Libraries were indexed using a NEBNext Singleplex primer set. The extent of sequencing ligation and library indexing was then quantified using a NEBNext Library Quant Kit for Illumina as per manufacturer's instructions. Quantified libraries were then combined at an equimolar ratio for next-generation sequencing using an Illumina MiSeq sequencer.

**Oxford nanopore library preparation and sequencing.** For nanopore sequencing via the Oxford Nanopore technology method, purified PCR amplified sequences containing encoded data were subjected to library preparation using the 1D Genomic DNA ligation sequencing kit (SQK-LSK109) from Oxford Nanopore Technologies following the manufacturer's protocols. Briefly, 0.5 pmol of the double-stranded DNA strands were used as starting material. The DNA was repaired and end-prepped using NEBNext FFPE DNA repair mix (NEB, M6630S) and NEBNext Ultra II End Repair/dA-Tailing Module (NEB, E7546) followed by bead purification using Agencourt AMPure XP beads (Beckman Coulter, A63880) at 1:2 sample to bead ratio. Adapters were then ligated to the end-prepped samples using the NEBNext Quick T4 DNA ligase (NEB, E6056S). The flow cells (R4.2.1) were primed, the sample was loaded onto the priming port of the flow cell and sequenced on the MinION, that generated ~500 K reads/hr. Sequencing was performed using the MinKNOW software (version 18.3.1, Oxford Nanopore Technologies) that converted raw data (in the form of fast5 files) into FastQ files which were used for downstream analysis.

**In silico sequence filtering for data recovery and decoding.** Digital data stored in DNA oligonucleotides was computationally recovered and decoded without previously knowing the composition of the original sequences. This was achieved by applying a series of filters to remove reads that contained synthesis errors or did not meet several initial boundary conditions. The initial boundary conditions included knowledge of the sequencing adapters used to prepared synthesis libraries for NGS, the barcodes used to identify oligonucleotide location and melody play order, the base composition of the surface anchored initiator oligonucleotide's 3′-terminus, the total number of oligonucleotides synthesized on the array in multiplex, and a rough estimation of the total number of base transitions needed to store all of the digital data. A total of three filters were used to clean-up the sequencing reads in order to stochastically estimate the composition of the original sequences without reference templates. Stochastic estimation was conducted using the Matlab built-in "seqlogo" function which graphically displays the sequence conservation at a particular position in the alignment of sequence (https://www. mathworks.com/help/bioinfo/ref/seqlogo.html). The first filter removed sequencing reads that did not contain the sequencing adapters used during NGS library preparation. The second filter removed sequencing reads that did not contain any instances of the unique locational barcodes. The third filter removed all sequencing reads that did not meet the expected number of base transitions. After the third filter, the composition of the sequences encoding digital data were determined from the remaining reads and then decoded into true sound played by the sinusoidal wave generator. To check if data recovery was successful, a fourth filter was used to remove any reads that did not match perfectly with the reference templates. A small or no drop in the total number of sequencing reads remaining at this point would indicate that stochastic estimation was largely successful after applying the third filter.

**Reporting summary.** Further information on research design is available in the Nature Research Reporting Summary linked to this article.

## Data availability
Raw sequencing data from Illumina and Oxford Nanopore MinION analysis of the final multiplex synthesis experiment is available on NCBI SRA under SRP262219. Source data are provided with this paper.

## Code availability
Encoding, NGS read filtering, stochastic estimation and decoding MatLab scripts used in this study are available at: https://github.com/dwiegand740/Photon_Enzymatic_Synthesis.

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

## Acknowledgements

The authors would like to thank Dr. Henry Lee for useful discussions regarding digital data storage, enzyme kinetics, and next-generation sequencing. The authors like to additionally thank Mr. Benjamin Vieira for the helpful discussion of musical nomenclature. K.G. is supported under Graduate Fellowships from the Fannie and John Hertz Foundation and the Charles Stark Draper Laboratory. This work was funded by the Wyss Institute for Biologically Inspired Engineering, Boston, Massachusetts.

## Author contributions

H.L., K.G., and G.M.C. conceptualized the project. H.L., D.J.W., K.G., S.P., and H.C. conducted the experiments involving: design and initiation of DNA synthesis (H.L., D.J.W., and K.G.), digital music encoding (H.L.), maskless photolithography system and flow cell development (H.L. and H.C.), and sequencing and analysis (H.L., D.J.W., and S.P.). H.L., D.J.W., S.P., R.E.K., and G.M.C. wrote the manuscript. H.L. and D.J.W. generated all figures; Fig. 4d–g include images generated in the BioRender software. R.E.K. and G.M.C. supervised the study.

## Competing interests

H.L., K.G., and G.M.C. have filed a patent application for the method described in this paper. The remaining authors declare no conflict of interest.
