## [Peer Review File · Nature Communications]

Reviewers' Comments:

Reviewer #1:

Remarks to the Author:

The authors of this paper demonstrate the enzymatic assembly of oligonucleotide DNA using a TdT with photolithography to control enzymatic polymerization. UV irradiation releases a cofactor from a photocleavable caging molecule starting synthesis and upon termination of UV irradiation polymerization is limited by the capture of the cofactor by an excess of caging molecules. While the processes used here are not independently novel the overall process as applied to DNA synthesis is novel and of interest to the field.

Because the arrest of polymerization is limited by diffusion speed after photoactivation the number of bases incorporated into the nascent strand are variable. This variable amount of base incorporation means the information encoded is read from the transition of one type of base to another rather than directly from the sequence. This distinction separates this type of synthesis from traditional chemical or enzymatic synthesis.

The authors demonstrate the utility of encoding information using this system with the synthesis to 12 oligos that each have 8 base transitions. This provides an acceptable proof of concept for basic feasibility, but it would be useful to understand the nature of the errors that occur during synthesis. Figure 4i shows the overall percentage of transition errors binned into deletion, insertion and mismatches but a more useful description of errors would be to show the prevalence of specific transition errors. For mismatches in particular the relation to neighboring oligo syntheses that indicates cross talk during the assembly would be useful to understand.

With the expansion of the data described above I recommend this manuscript for publication.

Reviewer #2:

Remarks to the Author:

This paper describes a multiplex enzymatic synthesis method where selective base addition is controlled by light-activated Co²⁺ ion uncaging and subsequent chelation with DMNP-EDTA. Light-controlled synthesis is not new and has been used in commercial DNA synthesis; what is new in this paper is the enablement of *enzymatic* synthesis using this control method. This is relevant because previous multiplex DNA synthesis chemistries did not use water as its main solvent, as is the case here. Multiplex synthesis is a requirement for scalable DNA data storage. This paper will definitely be interesting to researchers and practitioners in DNA data storage and is likely to generate further work in this area.

The paper has appropriately demonstrated that some level of control over base addition with UV light activation is possible. To do this, the authors have encoded a few notes from a song, and demonstrated that the intended sequences appear in a subset of the sequencing data coming from molecules collected from the array. This, by itself, already merits publication.

However, it does not completely make the case that the data can be successfully recovered in a DNA data storage scenario where the original sequences are unknown. To be clear, further improvement is possible and this review is not putting in doubt that this can be achieved, just that this is not demonstrated in this specific paper. Specifically, it looks like out of all the sequencing data, only a small percentage of the sequencing reads represents the intended sequences after computational filtering (6.6% in an Illumina sequencer and 0.06% in an ONT sequencer). Without an analysis of the remaining sequences, it is difficult to tell whether the data would be recoverable. This can be addressed in a couple of ways:

- Be clear that a claim on the ability to recover the data is not being made, and publish all the sequencing information and let other researchers do the analysis.

- The authors could do this analysis themselves; in particular, calculating the edit distance between the filtered out reads and the intended sequences, mapping these reads to the most likely intended sequence, and then analyzing the errors there would provide unparalleled insight. Further, the authors could show that the information can be recovered without the need of knowledge about the original sequences. In this case, the publication of all the sequencing information is strongly recommended so that other researchers can validate it.

Finally, the supplementary materials provide basic information for interested researchers to bootstrap experiments in the area, but it is still missing information that would allow for reproducibility of all results. In particular, the exact operations in a cycle are not provided along with the durations of each of these operations for each of the experiments, making them trial and error to reproduce. A table with all steps, their durations, for each of the experiments should be provided.

A few more specific comments:

- A demonstration with 12 unique sequences is far from "highly parallelized", as pitched in the abstract. This small scale demonstration is fundamental to scaling the technology and already a big accomplishment, so there is no need to overhype the results. Also, given that scaling it up will also be essential to its adoption in DNA data storage, the paper should provide a discussion on limits, potential roadblocks and roadmap to scaling the technology to higher levels of multiplexing.

- The sequences synthesized for this paper are short compared to those currently used for DNA data storage. The technology will only scale if the sequences can be longer. A discussion of limits, potential roadblocks and roadmap to scaling the number of bases in a synthesized sequence should also be provided.

- An additional important reference for durability of DNA is <https://onlinelibrary.wiley.com/doi/abs/10.1002/anie.201411378>.

- When commenting on scalability, it would be useful to see an estimation of the total power needed to maintain a certain synthesis capacity per period time rate, including optical and fluidics components.

- In Fig. 2c, how was base addition time verified? The paragraph that describes this experiment didn't really cover the "empirical" search for the proper illumination times for each transition in enough detail.

- The paper would benefit from a discussion that UV may cause DNA degradation. Based on available data and illumination times, the authors should at least estimate when the illumination may start affecting the integrity of the DNA being synthesized.

- In Sup. Fig. 1, the authors should separately list the diameter and both horizontal and diagonal center-to-center spot distances; it is not clear what the numbers on the right side represent. The scale for the microscope pictures is also missing.

- In Sup. Fig. 3, the 10% picture seems to show less worst confinement. It would be interesting to see comments from the authors on whether that is really the case and the reason.

- In Sup. Fig. 9a, the y axis of all figures is not readable.

- In Sup. Fig. 9c, it would help to divide the filtered out sequences by their closest edit distance

sequence and report them in the table as well. The same for Sup. Fig. 10c. Still about these two tables, it would be helpful to understand what is causing the filtering out of so many reads, and compare the two sequencing platforms and their effects on this.

Response to Reviewers

Photon-directed Multiplexed Enzymatic DNA Synthesis for Molecular Digital Data Storage

Authors: Howon Lee^{1,2}, Daniel J. Wiegand^{1,2}, Kettner Griswold^{1,2,3,4}, Sukanya Punthambaker^{1,2}, Honggu Chun⁵, Richie E. Kohman^{1,2,*}, George M. Church^{1,2,*}

Reviewer #1 (Remarks to the Author):

The authors of this paper demonstrate the enzymatic assembly of oligonucleotide DNA using a TdT with photolithography to control enzymatic polymerization. UV irradiation releases a cofactor from a photocleavable caging molecule starting synthesis and upon termination of UV irradiation polymerization is limited by the capture of the cofactor by an excess of caging molecules. While the processes used here are not independently novel the overall process as applied to DNA synthesis is novel and of interest to the field.

Because the arrest of polymerization is limited by diffusion speed after photoactivation the number of bases incorporated into the nascent strand are variable. This variable amount of base incorporation means the information encoded is read from the transition of one type of base to another rather than directly from the sequence. This distinction separates this type of synthesis from traditional chemical or enzymatic synthesis.

The authors demonstrate the utility of encoding information using this system with the synthesis to 12 oligos that each have 8 base transitions. This provides an acceptable proof of concept for basic feasibility, but it would be useful to understand the nature of the errors that occur during synthesis. Figure 4i shows the overall percentage of transition errors binned into deletion, insertion and mismatches but **a more useful description of errors would be to show the prevalence of specific transition errors**. For mismatches in particular the **relation to neighboring oligo syntheses that indicates cross talk during the assembly would be useful to understand**.

We thank the reviewer for the suggestion and agree that further insight into the nature of the synthesis errors beyond simple deletion, insertion and mismatch binning is warranted. We have therefore updated Figure 4 to show an additional graph describing the prevalence of these error types in the context of specific base transitions across the entire (3 x 4) array after multiplex synthesis and initial read filtering to subset those with sequencing adaptors. We had additionally revised the main text by updating **Figure 4i** to reflect the additional error analysis and removed:

A nominal portion of the filtered sequences contained errors despite our optimization efforts. These errors most likely arose from incomplete washing of the flow cell, unwanted crosstalk between spots (insertion 11.6% or mismatch 10.1%) or insufficient activation of TdT (deletion 26.1%) (Fig. 4i).

to add the new paragraph:

An appreciable percentage of the filtered reads contained errors; however, this did not hamper our ability to recover the correct oligonucleotide sequences from the sequencing data due to the physical redundancies built into our system. Errors could have arisen from insufficient activation of TdT, poor washing of the flow cell, and to much lesser extent, unwanted crosstalk between illumination spots. To characterize them, we categorized errors as either base transition insertions, deletions or mismatches. Single base deletions were found to be the most prevalent (25.8%), while a lower chance of single base insertions (13.4%) or mismatches (8.9%) was observed (Fig. 4i). Less than 5% of the filtered reads contained instances of two or more base transition errors. In breaking this down further, we did not find a strong correlation between any particular base transition (e.g. "A" to "T") and error type, although a noticeably high number of reads containing the transitions "G" to "C" and "C" to "A" were prone to error in general (Fig. 4i). This is in line with past observations where most base transitions involving "C" in some regard proved to be troublesome and required longer illumination times for sufficient enzymatic extension on the surface (Fig. 2c, Supplementary Fig. 5). Nevertheless, introducing logical redundancy such as error correction code^{3,13} and reducing the prevalence of base transition errors through further system optimization will be critical for the storage and recovery of high-quality digital data in DNA.

We additionally acknowledge the reviewer's inquiry into the relation between neighboring oligonucleotide syntheses in regards to mismatch errors. Currently, we believe it would be difficult to establish the specific effect of one particular synthesis spot on another without rigorous further experimentation but look forward to exploring this in greater detail in

the future. Undoubtedly, mismatch errors could have arisen from crosstalk between illumination spots; however, our optimization efforts to spatially confine synthesis reactions have greatly reduced this possibility (**Fig. 2b, Supplementary Fig. 1**). Further examination of specific base transition errors indicate a large percentage of total mismatches came from reads containing the “C” to “A” transition (**Fig. 4i**). The troublesomeness of this particular base transition may have contributed to an inflated mismatch error percentage, which is more of an issue with the kinetics of the enzyme rather than insufficient confinement and synthesis crosstalk.

With the expansion of the data described above I recommend this manuscript for publication.

Best Regards,

Adam Clore

Reviewer #2 (Remarks to the Author):

This paper describes a multiplex enzymatic synthesis method where selective base addition is controlled by light-activated Co²⁺ ion uncaging and subsequent chelation with DMNP-EDTA. Light-controlled synthesis is not new and has been used in commercial DNA synthesis; what is new in this paper is the enablement of *enzymatic* synthesis using this control method. This is relevant because previous multiplex DNA synthesis chemistries did not use water as its main solvent, as is the case here. Multiplex synthesis is a requirement for scalable DNA data storage. This paper will definitely be interesting to researchers and practitioners in DNA data storage and is likely to generate further work in this area.

The paper has appropriately demonstrated that some level of control over base addition with UV light activation is possible. To do this, the authors have encoded a few notes from a song, and demonstrated that the intended sequences appear in a subset of the sequencing data coming from molecules collected from the array. This, by itself, already merits publication.

However, it does not completely make the case that the data can be successfully recovered in a DNA data storage scenario where the original sequences are unknown. To be clear, further improvement is possible and this review is not putting in doubt that this can be achieved, just that this is not demonstrated in this specific paper. **Specifically, it looks like out of all the sequencing data, only a small percentage of the sequencing reads represents the intended sequences after computational filtering (6.6% in an Illumina sequencer and 0.06% in an ONT sequencer).**

Without an analysis of the remaining sequences, it is difficult to tell whether the data would be recoverable. This can be addressed in a couple of ways:

- Be clear that a claim on **the ability to recover the data is not being made**, and publish all the sequencing information and let other researchers do the analysis.
- The authors could **do this analysis themselves; in particular, calculating the edit distance between the filtered out reads and the intended sequences, mapping these reads to the most likely intended sequence, and then analyzing the errors there would provide unparalleled insight.**
- Further, the authors **could show that the information can be recovered without the need of knowledge about the original sequences.** In this case, the publication of all the sequencing information is strongly recommended so that other researchers can validate it

We thank the reviewer for bringing these concerns to our attention and providing a few different suggestions for us to address them. We have now published all of our sequencing data for the final multiplex synthesis experiment so that other researchers can perform further analysis on the data if desired. The revised manuscript includes an additional section entitled “Data Availability” which provides a link to our NCBI Sequence Run Archive upload. This SRA contains the sequence reads for both the Illumina MiSeq and Oxford Nanopore MinION runs.

When designing our encoding scheme, we sought to ensure that digital data could be recovered and decoded without previous knowledge of sequence composition or using additional error correction bits. This was achieved through physical redundancy (Ceze et al., 2019), where copies of the 12 oligonucleotide sequences synthesized in multiplex each existed on the array surface in large amounts at their pre-designated spots. To ascertain the sequences encoding data from the background noise using stochastic estimation, we needed to apply a stringent filtering process to remove those sequences with errors or erroneous information. The stringency of our filtering process was further compounded by the need to meet a set of initial boundary conditions specific to our system setup. As the reviewer pointed out, this resulted in only a small percentage of the total sequencing reads representing the intended sequences after computational filtering for the Illumina

and Oxford Nanopore data. Despite this, we were still able to recover and decode the digital data without knowing the composition of the sequences in both sequencing runs.

Inarguably, ensuring the recovery of high-quality data stored in DNA through physical redundancy may not be sustainable in the long term (Ceze et al., 2019). We believe that a drastic improvement could be made through the implementation of logical redundancies where highly stringent filtering may not be as necessary (Ceze et al., 2019; Lee et al., 2019). Our photon-directed enzymatic synthesis system is capable of handling sequences bearing logical redundancies, which will hypothetically increase the percentage of decodable reads. Since we wanted the primary focus of this manuscript to encompass the development of multiplexing in the context of enzymatic DNA synthesis, something that has not been well demonstrated before, we decided on a simpler digital data encoding scheme overall.

To further prove that we were able to recover digital data without knowledge of sequence composition, we have provided the reviewer with an overview of our data recovery and sequence filtering process. For our scheme, the initial boundary conditions included basic knowledge of the following:

1. The composition of the sequencing adaptors employed during library preparation (e.g. Illumina i7 adaptors).
2. The composition of the barcodes used to indicate musical play order and physical location of its sequence.
3. The total number of synthesized oligonucleotides on the array (e.g. 12 sequences).
4. The base composition of the surface anchored initiator oligonucleotide's 3'-terminus (e.g. string of "G").
5. A rough estimation of the total number of expected base transitions (barcode and sequence encoding information).

With these in place, our filtering process was as such:

1. **Load:** Raw sequencing data from NGS (e.g. Illumina MiSeq analysis)
 - a. Total reads obtained from the run was **216,252 (100%)**
2. **Filter #1:** Pre-screen raw sequencing data for the forward and reverse library adaptor sequences
 - a. Filtering using Smith-Waterman alignment algorithm from background noise (Smith & Waterman, 1981)
 - b. Now have a total of **100,760 reads (46.6%)**
3. **Filter #2:** Extract base transition information & generation subsets based on locational barcode sequences (index)
 - a. All data is stored in non-identical base transitions similar to previous work (Lee et al., 2019)
 - b. Extraction produces an array consisting of the type of transition & length of each homopolymer block
 - c. First 3 base transitions (including "G" initiator) are barcodes for physical location & melody play order
 - d. Remaining base transitions of subsetted sequences are unknown & variable; corresponds to stored data
 - e. Now have a total of **78,998 reads (36.5%)** split into 12 subsets based on the unique barcodes
 - f. Error analysis using templates, looking for base transitions deletions, insertions or mismatches (**Fig. 4i**)
4. **Filter #3:** Screen sequencing data for estimated number of base transitions in each subset & stochastic estimation
 - a. The estimated total number of base transitions is at least 3 due to the inclusion of the barcodes
 - b. Plotting these data on histogram shows the dominant population with 8 total base transitions
 - c. Now have a total of **16,741 reads (7.7%)** between all 12 subsets; largest percent drop
 - d. Stochastic estimation is now conducted to obtain the most probable strand sequence composition (**S9a**)
 - e. Sequence information can be converted into musical melody from here and played with wave generator
5. **Filter #4:** Final screen of filtered sequences for perfect matches
 - a. Determines the reliability of stochastic estimation to recover the original DNA sequences
 - b. This is the only time a comparison to the original DNA sequence templates is performed
 - c. Now have a total of **14,335 reads (6.6%)** consisting of perfect matches

In tracking the total sequencing reads remaining throughout our filtering process, it is abundantly clear that both **Filter #1** and **Filter #3** are the most significant in terms of percent filtered. For **Filter #1**, the dramatic loss in total reads could be attributed to poor post-synthesis NGS library preparation and/or an overall substandard sequencing run that resulted in high background noise. However, in the case of **Filter #3**, we observe that the majority of sequencing reads are removed

based on insufficiently meeting the initial boundary condition for estimated number of base transitions. This is in line with the error analysis performed on the sequencing reads after **Filter #2**, where we found single base transition deletions and insertions to be the most prevalent across the entire array during multiplex synthesis (**Fig. 4i, Supplementary Fig. X**). Overall, we believe that this analysis sufficiently satisfies the Reviewer’s inquiry regarding the reason for so little reads after filtering and the nature of errors found in those sequences that were filtered.

We have updated the following in the revised main text:

Filtering occurred in four discrete stages (Fig. 4h).

to read:

In silico filtering occurred in three discrete stages and was carried out to ensure that sequences synthesized in multiplex could be recovered without previous knowledge of their composition after fulfilling a set of predetermined initial boundary conditions (Fig. 4h, Supplementary Fig. 12, Methods).

and added the following to the same paragraph:

A fourth filter was then used to determine the effectiveness of stochastic estimation by comparing the composition of the dominant sequences to their reference templates. We observed only a small reduction in total remaining sequencing reads after the fourth filter indicating that the series of filtering process followed by stochastic estimation was largely successful (Supplementary Fig. 12).

and the following to the discussion:

We show that the first two measures of the “Overworld Theme” sheet music from the 1985 NES video game Super Mario Brothers™ can be encoded digitally into DNA and successfully recovered without previous knowledge of sequence composition despite the accumulation of some base transition errors.

We have additionally added the following figure to the supplementary materials as “Supplementary Figure 12”:

Supplementary Figure 12: A histogram indicating the distribution of non-identical base transitions for sequence reads after each filtering step from the Illumina MiSeq analysis of multiplex synthesis. **Filter #1** removes reads without sequencing adaptors from background noise (dark-blue bars). **Filter #2** removes sequencing reads without predetermined locational

barcodes (sky-blue bars). Barcodes consisted of 3 base transitions. **Filter #3** removes sequencing reads that did not fall within the estimated number of total base transitions (yellow). In this case, 7-8 base transitions were expected. After this filter, stochastic estimation of the unknown sequences occurs. **Filter #4** removes sequencing reads that were not perfect matches to reference sequences. Only a small difference between **Filter #3** and **Filter #4** was observed, indicating that decoding data from the oligonucleotides synthesized in multiplex can be done without previously knowing any original sequence composition.

Lastly, we have added the following to the methods section to better describe the filtering process:

In silico Sequence Read Filtering for Data Recovery and Decoding

Digital data stored in DNA oligonucleotides was computationally recovered and decoded without previously knowing the composition of the original sequences. This was achieved by applying a series of filters to remove reads that contained synthesis errors or did not meet several initial boundary conditions. The initial boundary conditions included knowledge of the sequencing adaptors used to prepared synthesis libraries for NGS, the barcodes used to identify oligonucleotide location & melody play order, the base composition of the surface anchored initiator oligonucleotide's 3'-terminus, the total number of oligonucleotides synthesized on the array in multiplex, and a rough estimation of the total number of base transitions needed to store all of the digital data. A total of 3 filters were used to clean-up the sequencing reads in order to stochastically estimate the composition of the original sequences without reference templates. The first filter removed sequencing reads that did not contain the sequencing adaptors used during NGS library preparation. The second filter removed sequencing reads that did not contain any instances of the unique locational barcodes. The third filter removed all sequencing reads that did not meet the expected number of base transitions. After the third filter, the composition of the sequences encoding digital data were determined from the remaining reads and then decoded into true sound played by the sinusoidal wave generator. To check if data recovery was successful, a fourth filter was used to remove any reads that did not match perfectly with the reference templates. A small or no drop in the total number of sequencing reads remaining at this point would indicate that stochastic estimation was largely successful after applying the third filter.

Finally, the supplementary materials provide basic information for interested researchers to bootstrap experiments in the area, but it is still **missing information that would allow for reproducibility of all results**. In particular, the **exact operations in a cycle are not provided along with the durations of each of these operations for each of the experiments, making them trial and error to reproduce**. A table with all steps, their durations, for each of the experiments should be provided.

We thank the reviewer for the suggestion. One advantage of our platform is the simplicity of standardizing the overall enzymatic synthesis process. A standard synthesis cycle is described in the material and methods of this manuscript, which represents the culmination of all experimental and optimization work. In order to ensure the highest level of reproducibility, the general order of operations did not significantly differ between each experiment. However, critical parameters such as illumination times and master mix composition certainly varied within the process to reach the optimal reaction conditions needed for multiplexed synthesis. We believe that the material and methods adequately describes this and that the addition of so many graphs would be potentially a source of confusion to readers. To provide some additional elaboration on the order of operations for a standard synthesis cycle, we have updated the text in the materials and methods to read:

Standard Synthesis Cycle

*A standard synthesis cycle consisted of loading the **required** mask image to the DMD, delivering 2 μL of synthesis master mix to the flow cell, and then applying UV irradiation to the bottom surface for 10-20 seconds depending on the base transition occurring at the individual illumination spots (**Fig. 2c**). For those base transitions that required very short illumination times, a minimum of 10 seconds of illumination was used and 20 seconds for longer cases (2-discrete step). A*

post-illumination incubation would then take place for at least 20 seconds *to ensure* optimal TdT extension and reaction quenching. Synthesis master mix was removed from the flow cell to the waste by vacuum *suction* and then washed with 20 μ L of 1X PBS. *Remaining 1X PBS wash was removed from the flow cell to the waste also by vacuum suction. To start the next cycle, a new mask image would be loaded into the DMD and this process was repeated as necessary for each nucleotide to be incorporated during a particular step in the case of multiplex synthesis (Fig. 3a,d,e, Supplementary Fig. 6).*

We have additionally added the following to the main text:

Based on this observation, the standard UV exposure protocol for multiplexed synthesis was determined as two normalized illumination times (Methods). For all synthesis spots ending with "C" in any synthesis cycle or step will be illuminated 20 seconds when they newly incorporate non-identical nucleotides at the 3' end, otherwise the UV exposure time was set to be 10 seconds uniformly.

A few more specific comments:

A demonstration with 12 unique sequences is far from "highly parallelized", as pitched in the abstract. This small scale demonstration is fundamental to scaling the technology and already a big accomplishment, so there is no need to overhype the results.

We thank the reviewer for pointing this out. We have removed the word "highly" from the abstract. We have additionally removed the word "highly" from the last sentence in the second paragraph of the main text of the revised manuscript.

Also, given that scaling it up will also be essential to its adoption in DNA data storage, the paper should provide a discussion on limits, potential roadblocks and roadmap to scaling the technology to higher levels of multiplexing.

We fully agree with the reviewer that scalability is essential for our technology to be adopted for DNA data storage applications. For higher levels of multiplexing, we will need to generate larger arrays consisting of smaller and denser synthesis spots. Because feature size on the array depends on parameters such as wavelength, mask resolution and projection optics, we hypothesize that this should be relatively easy to address by adapting commercially available high-resolution photolithography systems or time sharing scanning projections. However, confining the enzymatic reactions to smaller and more dense synthesis spots may require significant optimization (Namasivayam et al., 2003). We will need to adjust for the diffusion characteristics of the caging molecule, uncaged covalent cation and the enzyme as a function of flow-cell dimensionality, fluid viscosity and temperature in a larger scale system conducive for higher levels of multiplexing. Additionally, photon-directed DNA synthesis using smaller features may be more so affected by stray light cross-talk, inefficiencies in uncaging by UV irradiation, uncaged cation concentration distribution in the context of edge competition and the threshold of enzyme activity. Nevertheless, our optimization efforts and demonstration of multiplex synthesis on a smaller scale as presented in this manuscript will serve as an effective model for mapping out the road to scaling up.

The sequences synthesized for this paper are short compared to those currently used for DNA data storage. The technology will only scale if the sequences can be longer. A discussion of limits, potential roadblocks and roadmap to scaling the number of bases in a synthesized sequence should also be provided.

We thank the reviewer for the suggestion and fully agree that the ability to synthesize oligonucleotides with longer lengths will be critical for maximizing our capacity to store data in DNA. The multiplexed synthesis demonstration reported in this manuscript was a manual process, so the implementation of automation for the delivery of reaction reagents, flow-cell washing and synchronization of photonics will allow for the streamlined production of longer oligonucleotides. However, we expect that as we increase the length of synthesized oligonucleotides, new challenges will undoubtedly present themselves. For instance, it is well-known that single-stranded DNA oligonucleotide sequences will form secondary

structures under thermodynamically favored conditions in an aqueous-based system (Bosco et al., 2014). This could result in the 3'-terminus of some oligonucleotides on the array to be inaccessible by enzyme, making it difficult for the next nucleotide extension step to be completed. While secondary structure is less of an issue with chemical synthesis techniques due to the presence of organic solvents, our enzymatic synthesis can be supplemented with additives that normalize or enhance extension despite difficult secondary structure. For example, the addition of a DNA binding protein (Shishmarev et al., 2014), either in the bulk reaction or fused to TdT without diminishing its activity, could be implemented on the road to unrestricted production of long oligonucleotides.

We have modified and expanded the following sentence to address scalability in terms of multiplexing and the synthesis of longer oligonucleotides:

While enzyme-based oligonucleotide synthesis technology is still in its infancy, we envision that bridging overall cleaner and faster DNA oligonucleotide synthesis with cutting-edge, high-density array photolithographic techniques will pave the way for its widespread adoption and further promote practical molecular data storage technology.

to read:

While enzymatic-based DNA oligonucleotide synthesis technology is still in its infancy, we envision that pairing a overall cleaner, faster and more flexible synthesis methodology with high-density array photolithographic techniques will pave the way for its widespread adoption. However, in order to achieve this, there are still several remaining challenges that need to be first overcome. For example, having the ability to scale easily in terms of both higher levels of multiplexing and the synthesis of longer DNA oligonucleotide sequences (>100-nt) will be essential in promoting the further use of our photon-directed synthesis platform as practical molecular data storage technology. We believe that the synthesis of a greater number of oligonucleotides in multiplex will be enabled by utilizing larger arrays with denser patterning. Because feature size on the array largely depends on parameters such as wavelength, mask resolution and projection optics, this should be relatively easy to address through the implementation of commercially available high-resolution photolithography systems. Nevertheless, confining the enzymatic reactions to smaller and more dense synthesis spots may require significant optimization and the overall process could be affected in a greater capacity by improper diffusion of DMNP-EDTA, stray light cross-talk or unforeseen inefficiencies in uncaging by patterned UV irradiation²¹. Similarly, achieving longer oligonucleotide synthesis with our system requires the implementation of full automation and may be affected by phenomenon such as the formation of secondary structure that makes the 3'-terminus of the DNA inaccessible by the enzyme without intervention by DNA binding proteins or elevated reaction temperatures^{22,23}. This limitation is an inherent problem with aqueous-based reaction systems, which is both well-recognized and currently being addressed with success by several other groups^{12,24}.

An additional important reference for durability of DNA is
<https://onlinelibrary.wiley.com/doi/abs/10.1002/anie.201411378>.

We thank the reviewer for providing this important reference. We have added the citation to the introductory paragraph of the revised manuscript and have updated our reference numbering.

When commenting on scalability, it would be useful to see an estimation of the total power needed to maintain a certain synthesis capacity per period time rate, including optical and fluidics components.

We appreciate the reviewer's suggestion. We estimate approximately 30 μJ /base-transition of optical power consumption based on a power requirement of $2\text{W}/\text{cm}^2$, spots with a 100 μm diameter and an average of 10 seconds of illumination per base transition. To synthesize the 12 oligonucleotides in multiplex, a total of 87 base transitions were needed to encode 110 bits of digital data. This yields approximately 23.7 $\mu\text{J}/\text{bit}$. Because we performed multiplex synthesis with manual pipetting for this study, it is currently not possible to estimate the power consumption of fluidic components.

In Fig. 2c, how was base addition time verified? The paragraph that describes this experiment didn't really cover the "empirical" search for the proper illumination times for each transition in enough detail.

We thank the reviewer for the inquiry. Our empirical search was conducted by testing four discrete illumination times for each possible base transition. A full overview of all tested conditions is provided in a new supplementary figure and caption, which is properly referenced in the revised main text:

C→A	T→A	G→A	A→T	G→T	C→T	A→G	T→G	C→G	A→C	T→C	G→C
10s	4s	4s	4s	4s	10s	4s	4s	8s	8s	8s	8s
12.5s	6s	6s	6s	6s	12.5s	6s	6s	10s	10s	10s	10s
15s	8s	8s	8s	8s	15s	8s	8s	12.5s	12.5s	12.5s	12.5s
17.5s	10s	10s	10s	10s	17.5s	10s	10s	15s	15s	15s	15s

Supplementary Figure 5: Verification of the based addition time. The base addition time for each possible base transition was empirically determined by testing four discrete illumination time followed by fluorescence imaging via splint-end ligation of a probe sequence containing a 3'-Cy3 fluorophore. We chose the base addition time of each case based on the fluorescence signal intensity and spot confinement at the same time. As expected, most "C" involving additions tends to require more illumination time than others.

The paper would benefit from a discussion that UV may cause DNA degradation. Based on available data and illumination times, the authors should at least estimate when the illumination may start affecting the integrity of the DNA being synthesized.

We thank the reviewer for the suggestion. Patterned UV light at a wavelength of 365 nm is used to illuminate the array surface for the initiation of each uncaging and base addition step. We suspect that prolonged exposure may become problematic in cases of longer synthesis (>50 transitions); however, this is unlikely, as it is well established that UV-A radiation (315-400 nm) is a poor inducer of DNA damage (Rastogi et al., 2010). Furthermore, some previous iterations of light directed oligonucleotide synthesis using chemical methods have not explicitly reported DNA damage to be a major limiting factor of the technology (Gao et al., 2001; Pease et al., 1994; Singh-Gasson et al., 1999). It is important to note, however, that these authors were conscientious of the adverse effects of UV light on nucleic acids by utilizing photochemistry compatible with wavelengths in the UV-A spectrum or greater (Singh-Gasson et al., 1999).

In Sup. Fig. 1, the authors should separately list the diameter and both horizontal and diagonal center-to-center spot distances; it is not clear what the numbers on the right side represent. The scale for the microscope pictures is also missing.

We thank the reviewer for pointing this out. We have updated the axes so that intended information is correctly portrayed in the figure. This is reflected in the caption description. Images of each post-extension DMD pattern were captured using a Typhoon FLA 9000 Imager and are shown with the 1 mm scale bar. We have added the requested information regarding spot distances to the figure. The following text was added to description of Supplementary Fig. 1:

“Images were captured and analyzed using a Typhoon FLA 9000 Imager. The y-axis indicates the diameter of the circular spots and mask used to generate the illumination pattern. The x-axis indicates the total concentration DMNP-EDTA in the reaction mixture. Additionally, a diagram showing the center-to-center vertical, diagonal and horizontal distances between spots is provided.”

In Sup. Fig. 3, the 10% picture seems to show less worst confinement. It would be interesting to see comments from the authors on whether that is really the case and the reason.

We thank the reviewer for the inquiry and agree that spots in the picture depicting a synthesis master mix containing 4U/ μ L TdT and 10% glycerol appear to be “blotchy” and thus, less confined. We believe that this appearance may have been caused by increased diffusion of the cofactor ion due to the lower amount of glycerol present in the reaction mix, which resulted in higher TdT activity and ssDNA product on the surface. Nevertheless, the clear absence of signal between the individual spots suggests acceptable confinement as compared to the poorer cases indicted in Supplementary Figure 1.

In Sup. Fig. 9a, the y axis of all figures is not readable.

We thank the reviewer for pointing this out. We have updated the figure with higher quality images so that the y-axis is legible.

- In Sup. Fig. 9c, it would help to divide the filtered out sequences by their closest edit distance sequence and report them in the table as well. The same for Sup. Fig. 10c. Still about these two tables, it would be helpful to understand what is causing the filtering out of so many reads, and compare the two sequencing platforms and their effects on this.

We thank the reviewer for the suggestion and believe that it is addressed in the above comments. In addition, we have provided an update to Figure 4i and added Supplementary Figure 12, which shed more light on synthesis errors and the stringent filtering process carried out to ascertain the composition of the synthesized sequences.

References

- Bosco, A., Camunas-Soler, J., & Ritort, F. (2014). Elastic properties and secondary structure formation of single-stranded DNA at monovalent and divalent salt conditions. *Nucleic Acids Research*, *42*(3), 2064–2074.
- Ceze, L., Nivala, J., & Strauss, K. (2019). Molecular digital data storage using DNA. *Nature Reviews. Genetics*, *20*(8), 456–466.
- Gao, X., LeProust, E., Zhang, H., Srivannavit, O., Gulari, E., Yu, P., Nishiguchi, C., Xiang, Q., & Zhou, X. (2001). A flexible light-directed DNA chip synthesis gated by deprotection using solution photogenerated acids. *Nucleic Acids Research*, *29*(22), 4744–4750.
- Lee, H. H., Kalhor, R., Goela, N., Bolot, J., & Church, G. M. (2019). Terminator-free template-independent enzymatic DNA synthesis for digital information storage. *Nature Communications*, *10*(1), 2383.
- Namasivayam, V., Larson, R. G., Burke, D. T., & Burns, M. A. (2003). Light-induced molecular cutting: localized reaction on a single DNA molecule. *Analytical Chemistry*, *75*(16), 4188–4194.
- Pease, A. C., Solas, D., Sullivan, E. J., Cronin, M. T., Holmes, C. P., & Fodor, S. P. (1994). Light-generated oligonucleotide arrays for rapid DNA sequence analysis. *Proceedings of the National Academy of Sciences of the United States of America*, *91*(11), 5022–5026.
- Rastogi, R. P., Richa, Kumar, A., Tyagi, M. B., & Sinha, R. P. (2010). Molecular mechanisms of ultraviolet radiation-induced DNA damage and repair. *Journal of Nucleic Acids*, *2010*, 592980.
- Shishmarev, D., Wang, Y., Mason, C. E., Su, X.-C., Oakley, A. J., Graham, B., Huber, T., Dixon, N. E., & Otting, G. (2014). Intramolecular binding mode of the C-terminus of Escherichia coli single-stranded DNA binding protein determined by nuclear magnetic resonance spectroscopy. *Nucleic Acids Research*, *42*(4), 2750–2757.
- Singh-Gasson, S., Green, R. D., Yue, Y., Nelson, C., Blattner, F., Sussman, M. R., & Cerrina, F. (1999). Maskless fabrication of light-directed oligonucleotide microarrays using a digital micromirror array. *Nature Biotechnology*, *17*(10), 974–978.
- Smith, T. F., & Waterman, M. S. (1981). Identification of common molecular subsequences. *Journal of Molecular Biology*, *147*(1), 195–197.

Reviewers' Comments:

Reviewer #1:

Remarks to the Author:

The response from the authors to my questions and to the other reviewers questions satisfy me. I recommend this manuscript for publication without reservation.

Reviewer #2:

Remarks to the Author:

This reviewer really appreciates the authors' effort and care in answering questions. The only remaining suggestion is to add the stochastic estimation algorithm or a pointer to the algorithm/paper used in supplemental materials to ensure reproducibility.

Response to Reviewers

Photon-directed Multiplexed Enzymatic DNA Synthesis for Molecular Digital Data Storage

Authors: Howon Lee^{1,2}, Daniel J. Wiegand^{1,2}, Kettner Griswold^{1,2,3,4}, Sukanya Punthambaker^{1,2}, Honggu Chun⁵, Richie E. Kohman^{1,2,*}, George M. Church^{1,2,*}

Reviewer #1 (Remarks to the Author):

The response from the authors to my questions and to the other reviewers questions satisfy me. I recommend this manuscript for publication without reservation.

Adam Clore

Dear Adam, we are sincerely grateful for your time as well as insightful input and suggestions. We thoroughly enjoyed answering your questions and believe that the expanded error analysis improves the manuscript significantly.

Reviewer #2 (Remarks to the Author):

This reviewer really appreciates the authors' effort and care in answering questions. The only remaining suggestion is to add the stochastic estimation algorithm or a pointer to the algorithm/paper used in supplemental materials to ensure reproducibility.

We thank the reviewer for the kind words. We really enjoyed answering your questions and believe that the manuscript has been dramatically improved because of them. We have addressed your last suggestion by uploading the sequence filtering and stochastic estimation Matlab code to our Github repository linked in the “Code Availability” section as well as adding the following text with the appropriate link to the methods section of the manuscript:

Stochastic estimation was conducted using the Matlab built-in “seqlogo” function which graphically displays the sequence conservation at a particular position in the alignment of sequence (<https://www.mathworks.com/help/bioinfo/ref/seqlogo.html>).